# Particular Morphological Features in the Diagnosis of Pediatric *Helicobacter pylori* Gastritis: A Morphometry-Based Study

**DOI:** 10.3390/jcm9113639

**Published:** 2020-11-12

**Authors:** Ana-Maria Teodora Domșa, Dan Gheban, Camelia Lazăr, Bogdan Pop, Cristina Maria Borzan

**Affiliations:** 1Department of Pathology, “Iuliu Hațieganu” University of Medicine and Pharmacy, 400012 Cluj-Napoca, Romania; domsa_dora@yahoo.com (A.-M.T.D.); dgheban@gmail.com (D.G.); pop.bogdan21@gmail.com (B.P.); 2Emergency Clinical Hospital for Children, 400370 Cluj-Napoca, Romania; 3Department of Pathology, the Oncology Institute ‘Prof. I. Chiricuta’, 400015 Cluj-Napoca, Romania; 4Department of Public Health and Management, “Iuliu Hațieganu” University of Medicine and Pharmacy, 400012 Cluj-Napoca, Romania; borzancristina@yahoo.com

**Keywords:** *Helicobacter pylori*, gastritis, children, morphometry, histopathology, foveolar hyperplasia

## Abstract

Background: Current pediatric guidelines recommend the use of the Updated Sydney Classification for gastritis to assess histological changes caused by *Helicobacter pylori* (*H. pylori*) infection. The purpose of this study was to investigate the morphometric alterations of the antral mucosa in relation to pediatric *H. pylori* infection. Methods: A total of 65 cases were considered eligible. Apart from scoring the biopsies according to the recommendations, foveolar hyperplasia (FH) was assessed. The following measurements were performed on digital slides: total mucosal thickness, foveolar and glandular length, number of glandular cross sections per 40X field, glandular diameter, and distance between glands. Results: The thickness of the antral mucosa increased along with the bacterial density and the intensity of inflammation in *H. pylori*-infected children (*p* < 0.05). FH was significantly associated with the presence of *H. pylori* (*p* < 0.001) and also exhibited a greater length of the foveolar and glandular structures and an increased glandular diameter (*p* < 0.05), but without influencing the thickness of the mucosa. Conclusions: Our results reinforce the fact that FH is not only an important histologic characteristic of gastropathy, but is also a significant change observed in *H. pylori* infection in children and may be considered for reporting when evaluating pediatric gastric biopsies.

## 1. Introduction

*Helicobacter pylori* (*H. pylori)* infection represents a major public health problem worldwide, still being the most frequent chronic infection in both adults and children. It is estimated that more than 50% of the global population is affected [1].

The infection occurs mainly in childhood; in developing countries, 50% of children are infected by the age of five [2]. Bacterial colonization persists throughout life in the absence of adequate treatment [3].

*H. pylori* is considered the most important risk factor in the mechanism of gastric carcinogenesis [4]. A universally accepted cascade of precancerous lesions has been described [5]. The progression of the lesions is dynamic and individual-dependent [6]. It is also known that the disease outcome is influenced by the age at acquisition of the infection [7].

When considering the diagnostic strategies of *H. pylori* infection in children, current pediatric guidelines support upper gastrointestinal endoscopy with biopsies as the most reliable method [8]. They also recommend the use of the Updated Sydney Classification for gastritis [9] for the assessment of histopathological changes [8].

While in adults the symptomatology of the infection is rather specific, most pediatric patients exhibit non-specific symptoms [10]. When compared to adults where histopathological changes are well defined, the microscopical alterations observed in pediatric patients are less conclusive and sometimes conflicting [11,12].

Although there are several morphometric studies analyzing the gastric mucosa of *H. pylori*-infected adults, data regarding pediatric patients are limited [13,14]. Taking into consideration that the clinical and paraclinical features encountered in children may differ from those observed in adults, we aimed to investigate the morphometric characteristics of the gastric antral mucosa in relation to pediatric *H. pylori* infection.

## 2. Materials and Methods

We performed a retrospective study on 93 consecutive cases of pediatric patients aged 0 to 18 years. The examined biopsies resulted from upper gastrointestinal endoscopy (UGE) performed at the Emergency Clinical University Hospital for Children in Cluj-Napoca between 1 January 2017 and 31 December 2017. For the purpose of investigating various digestive symptoms, three biopsy specimens were collected from the antral region, the body of the stomach, and the duodenum. Exclusion criteria consisted of age older than 18 and medical history of *H. pylori* infection. No selective criteria were applied in relation to the patients’ symptoms.

An experienced pathologist appraised all of the available slides originating from the antral mucosa. The scoring of the biopsies was assessed according to the Updated Sydney Classification system. *H. pylori* infection status was based on hematoxylin–eosin and Giemsa stains; the patient was considered infected if both stains were positive. Additionally, we recorded foveolar hyperplasia (FH), described as elongation along with a twisted or branching aspect of the foveola, as proposed by previous studies [9,15], and we graded it into absent, mild, or marked, based on the visual aspect.

The morphometric technique was based on a previously developed method used to evaluate adult gastric biopsies [13]. In order to perform objective measurements, we considered the samples adequate if they had at least one 20X microscopic field displaying the entire thickness of the mucosa, with the mucosal surface perpendicular to the cutting plane. A total of 65 cases met the above-mentioned criteria and were included in the final analysis.

The glass slides of the antral biopsies were digitized with the aid of a slide scanner (Pannoramic SCAN II, 3DHISTECH, Budapest, Hungary) using 20X objective magnification and visualized using the Case Viewer software (3DHISTECH, Budapest, Hungary). According to the previously described protocol [13], two other pathologists, who were unaware of the study variables, performed the following measurements: total mucosal thickness, foveolar length, glandular length, number of glandular cross sections per 40X field, glandular diameter, and distance between glands. The average between the two sets of measurements was used for the analysis. Based on the obtained averages, we also calculated the following ratios: foveolar length/total mucosal thickness, glandular length/total mucosal thickness, and foveolar length/glandular length.

The study was conducted in accordance with the Declaration of Helsinki and the protocol was approved by the ethical committee of the University of Medicine and Pharmacy Cluj-Napoca (208/16 May 2017). Informed consent was obtained prior to UGE from the parents or legal guardians of the subjects.

Anthropometric, clinical, histological, and morphometric data were collected anonymously.

### Statistical Analysis

The statistical analysis was performed using R and R Commander version 3.6.2 and Microsoft Office 365 ProPlus. Continuous data with normal distribution are described as mean ± SD, and continuous data without normal distribution are described as median and interquartile range (IQR). Nominal variable data are shown as percentages. The normality was tested using the Shapiro­–Wilk test. Homogeneity of variance was tested using Bartlett’s test. The differences between groups were assessed using Student’s *t*-test or one-way ANOVA for continuous variables with normal distribution, and the Mann–Whitney U test or Kruskal–Wallis test for continuous variables without normal distribution. In case of heteroscedasticity, Welch’s *t*-test or Welch’s ANOVA were used. For all pairwise comparisons, the Tukey–Kramer test was used after a significant one-way ANOVA, and the Games–Howell test after a significant Welch’s ANOVA. Fisher’s exact test of independence and Pearson’s Chi-squared test of independence were used for proportions. Values of *p* < 0.05 were considered significant, with a 95% confidence level for intervals. To control the familywise error rate in post-hoc tests when using multiple pairwise comparisons, critical *p*-values were Bonferroni-corrected.

## 3. Results

A total of 65 pediatric patients with a median age of 14 years [11,12,13,14,15,16,17] were considered eligible for this study. As shown in Table 1, no age difference was found between the two genders (*p* > 0.05).

*H. pylori* infection was detected in 34 (52.3%) of the 65 patients analyzed. In the *H. pylori*-positive group, 10/34 were male, while in the *H. pylori*-negative group, 11/31 were male, without any significant age- or sex-related differences (*p* > 0.05).

In the *H. pylori*-negative group, 23/31 patients presented no inflammation, while the remaining eight presented mild inflammatory infiltrate. In the *H. pylori*-positive group, 9/34 presented mild inflammatory infiltrate, 13/34 moderate inflammatory infiltrate, while the remaining 12 patients presented severe inflammatory infiltrate.

Seventeen cases showed atrophy, of which 15 (88.23%) were graded as mild atrophy and none as severe atrophy; of these patients, 11 had *H. pylori* infection. No relationship could be established between the presence of atrophy and *H. pylori* positivity or the age of the patients (*p* > 0.05).

When evaluated by the visual scale, FH was observed in 39 (60%) cases. The proportion of patients who presented branching or tortuosity of the foveola was significantly higher in the *H. pylori*-positive group, with 28 out of 34 *H. pylori*-positive patients (82.4%) exhibiting FH (*p* = 0.0003). The relationship between the presence of *H. pylori* and the extent of FH is displayed in Figure 1 (*p* = 0.002). In addition, the probability that a subject with FH was indeed *H. pylori*-positive was 76% (95%CI, 60–89%), whereas the absence of FH excluded *H. pylori* infection in 71% (95%CI, 60–79%) of the cases.

Furthermore, we observed an increase in the tortuosity and branching of the foveola along with the increase in chronic inflammatory infiltrate (*p* = 0.02), as illustrated in Figure 2.

As shown in Figure 1, there was no significant difference between the extent of FH based on the visual scale and the presence of atrophy (*p* > 0.05). Tortuosity or branching was present in 26/48 cases without atrophy. Out of the 17 atrophic cases, eight (47.1%) presented mild and five (29.4%) marked FH. The presence of FH was not influenced by the age or sex (*p* > 0.05) of the patients.

Table 2 summarizes the morphometric parameters determined from the antral mucosa.

The presence of *H. pylori* infection significantly altered the total mucosal thickness. The infected patients had a thicker mucosa than the noninfected group (*p* = 0.0007). Figure 3 details the relationship between the mucosal thickness and *H. pylori* density (*p* = 0.003).

When comparing *H. pylori* density with the glandular length, we observed an increase in the length of the glands along with the bacterial colonization grade (*p* = 0.004), as shown in Figure 3. Moreover, the glandular length/total mucosal thickness ratio was also influenced by *H. pylori* density (*p* = 0.02), as the ratio increased with the severity of bacterial colonization.

The foveolar length of *H. pylori*-positive patients was greater than the foveola of *H. pylori*-negative patients and the length increased with *H. pylori* colonization grade (*p* = 0.045).

The glandular diameter was increased in *H. pylori*-infected patients (*p* = 0.03); the mean diameter of the glandular structures in the negative group was 36.79 µm, whereas the mean diameter of the positive group was 38.66 µm.

Regarding the association between the morphometric measurements of the antral mucosa and the extent of FH, as evaluated based on the visual scale, we noted a significant increase in the foveolar length (*p* = 0.03), as well as an increase in the length of the glands (*p* = 0.03), with increasing degree of FH (Figure 4). However, despite these differences, no significant relationship could be established between the extent of FH and the three determined ratios.

Depending on the degree of FH, an increase in the glandular diameter was observed (*p* = 0.003) together with a decrease in the number of glands/40X (*p* = 0.007).

None of the other morphometric variables were significantly influenced by the degree of FH.

The only two morphometric parameters that were significantly altered by the presence of atrophy, as graded according to the visual scale, were the diameter of the glandular structures (*p* = 0.03) and distance between glandular structures (*p* = 0.003); Figure 5 reports these changes in regard to the extent of atrophy.

Surprisingly, no relationship was identified between the presence of atrophy and the number of glands/40X. Furthermore, atrophy did not alter the mucosal thickness, foveolar length, glandular length, or the ratios between the three parameters, irrespective of the grade (*p* > 0.05).

When analyzing the intensity of the chronic inflammation in relation to the morphometric parameters, we observed that the mucosal thickness and the length of glandular structures significantly increased along with the intensity of the inflammatory infiltrate (*p* = 0.0007 and *p* = 0.001). These changes are illustrated in Figure 6.

We also noted a decrease in the number of glandular structures/40X (*p* = 0.014), from a mean of 18.73 ± 3.95 in the absence of inflammation to a mean of 15.45 ± 2.90 for the cases that presented severe chronic inflammatory infiltrate.

None of the other morphometric parameters were significantly influenced by the degree of the inflammatory infiltrate (*p* > 0.05).

## 4. Discussion

In this study, we evaluated the histologic changes of pediatric antral mucosa as a result of *H. pylori* infection. As recommended by current guidelines, the grading of gastritis in pediatric patients is based on visual analogue scales that are also used in adults [8]. Our study provides morphometric documentation of the analyzed parameters through a set of objective measurements.

Considering that in both in adult and pediatric patients, characteristic changes are better represented at the antral level [9,16], we limited our evaluation to antral biopsies.

Notably, our study found that in *H. pylori*-infected children, the thickness of the antral mucosa increased dependent on the bacterial density (*p* < 0.01) and on the intensity of the inflammatory infiltrate (*p* < 0.05), regardless of the age of the patient. According to previous studies, the inflammatory pattern differs in children compared to adults, the former presenting predominantly chronic inflammation, with reduced signs of activity [17,18,19]. Our results showed that enhancement of the chronic infiltrate was associated with thickening of the mucosa and elongation of the glandular structures along with a decrease in the number of glands, but the foveolar length did not change significantly. This particular set of changes could suggest that the inflammatory infiltrate is no longer predominantly located in the superficial part of the mucosa, centered around the gastric pits, but that it has already advanced toward the deeper regions of the mucosa, in the secretory area; according to the classification of gastritis, these changes fall into the chronic gastritis group [9].

FH is described as increased length and tortuosity of the foveola associated with the expansion of the proliferative compartment [9]. Although it is generally acknowledged that FH is a characteristic finding in reactive gastropathy, uncertainty persists regarding the connection between *H. pylori* and FH [20]. Early studies stated that the corkscrew appearance of the foveola is observed in all types of gastritis [21] and that FH arises either as a compensatory reaction to increased cell exfoliation or as a response to inflammatory mediators [22]. Additionally, in reactive gastropathy, the diagnosis should be based on a constellation of changes (FH together with edema and hyperemia of lamina propria and prominent interfoveolar smooth muscle fibers, in the absence of significant inflammation), none of these being specific if taken individually. Furthermore, previous investigators agree that a moderate or severe inflammatory infiltrate of the gastric mucosa favors the diagnosis of *H. pylori* infection, and if prominent FH is present, the patient probably associates a chemical gastropathy component [21].

Based on existing criteria from previous research, we evaluated FH both in a semiquantitative manner and by using morphometric parameters. We found that, when appraised by the visual scale, FH was significantly associated with the presence of *H. pylori* (*p* < 0.001) and also exhibited a greater length of the foveolar and glandular structures (*p* < 0.05), supporting the concordance between the somewhat subjective visual assessment of the pathologist and the objective measurements. Our results are in agreement with a previous investigation, which stated an increase in foveolar length as being significantly associated with *H. pylori* infection in children [14]. Moreover, FH became more pronounced as the degree of chronic inflammatory infiltrate increased (*p* < 0.05). This change is supported by previous reports stating that hyperplasia may be seen in all forms of gastritis [9]. Associated with FH, we also noticed an augmented diameter of the glandular structures, but without significantly altering the thickness of the mucosa. This observation is supported by Genta, who described a relatively normal or a somewhat hypertrophic glandular component underlying FH in patients with chemical gastropathy [21].

Atrophy of the gastric mucosa is defined as loss of glandular structures, which become sparse and small, leading to thinning of the mucosa. In adults, a decrease in the mucosal thickness is usually observed with increasing atrophy [9].

Interestingly, in our study, the decrease in mucosal thickness and number of glands was not objectified in the cases in which atrophy was reported (*p* > 0.05). Another important observation was that the distance between the glands increased in cases with atrophy, regardless of the degree of inflammation (*p* < 0.01). It should be noted that most of the cases were graded as mild atrophy. As emphasized in the literature, the loss of normal glandular structures is the first specific recognizable step in the precancerous cascade, usually as the result of a prolonged inflammatory process; the transition from non-atrophic gastritis, characterized by increased mononuclear leukocytic infiltration in the lamina propria and well-preserved glands, to atrophic gastritis, diagnosed when loss of glandular structures is identified, extends over several years [23].

Given these discrepancies, we believe that in pediatric patients, we should not rely only on visual scales, but also add basic morphometric measurements to our evaluation, especially in selected cases with mild atrophy, where reporting is extremely variable depending on examiners [24].

Unlike atrophy, FH seems to be one of the early changes associated with *H. pylori* infection in children; we observed that FH decreases in intensity as atrophy occurs. Future work is needed to determine if FH may be somewhat meaningful in determining the duration of the infection and if it is associated with the presence of specific virulence factors in the infecting *H. pylori* strain, known to be a determinant for the outcome of the infection.

To the best of our knowledge, this is the first study that extensively evaluated the antral mucosa of *H. pylori*-infected pediatric patients through multiple objective measurements performed both in the superficial area of the antral mucosa and at the level of the deep glandular component, and that sought to establish links between morphometric parameters and the visual analog scale used for the grading of gastritis.

The present study has several limitations. The research was conducted in a retrospective manner, so we had limited data on previous *H. pylori* eradication therapy and patients’ associated pathologies and medication regimens. Moreover, the study was conducted in a single hospital-based center, with a short time span and limited number of cases. The latest pediatric guidelines for the management of *H. pylori* recommend at least six gastric biopsies to be collected for the diagnosis; thus, by collecting a smaller number of biopsies, the infection could be underdetected. Although it is recognized that the most significant changes are encountered in the antrum, the changes in the corpus would have been worth evaluating.

The age of the analyzed patients was non-normally distributed, with only nine patients being under 10 years old, but this aspect could be explained by the sparse symptomatology found in children and because acquisition of the infection increases with age.

We think that more morphometric-based studies would provide support for pathologists in establishing the most appropriate histological grading according to the visual analogue scales and have the potential to identify additional parameters that can act as prognostic or progression factors. Prospective studies, comprising a significant number of pediatric patients, could also evaluate the dynamics between tortuosity and atrophy, along with age and infection progression. This could help in identifying patients at risk of developing more severe lesions and could subsequently lead to the tailoring of diagnostic and treatment methods according to age.

## 5. Conclusions

Our study supports the fact that FH is not only a significant change in gastropathy, but is also significantly associated with *H. pylori* infection in children. For routine diagnosis, it may not be feasible to measure glandular and foveolar lengths, but there seems to be a satisfactory correspondence between the visual analysis of FH and morphometry, and thus it may be worth adding it to the evaluation in pediatric patients.

In children, *H. pylori* infection associates an increase in thickness of the antral mucosa and in glandular length, in contrast to adults where chronic atrophic gastritis, meaning loss of glands and thinning of the mucosa, is a common change in long-term *H. pylori* infection.

As the symptomatology of *H. pylori* infection is different in children compared to adults, careful evaluation is required when extrapolating the histological characteristics of the infection from adults to children. An adaptation of the classification used in adults, with the addition of parameters that could act as prognostic factors for the evolution of the disease in children, would likely be welcomed.

## Figures and Tables

**Figure 1 jcm-09-03639-f001:**
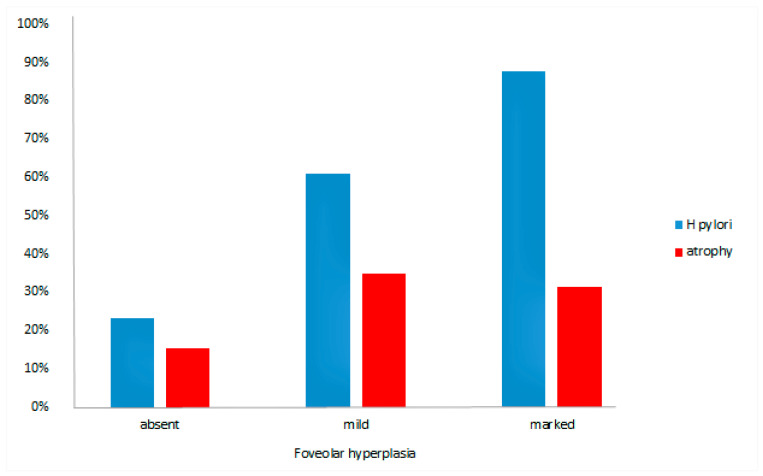
The degree of foveolar hyperplasia depending on the presence of *H. pylori* infection and atrophy.

**Figure 2 jcm-09-03639-f002:**
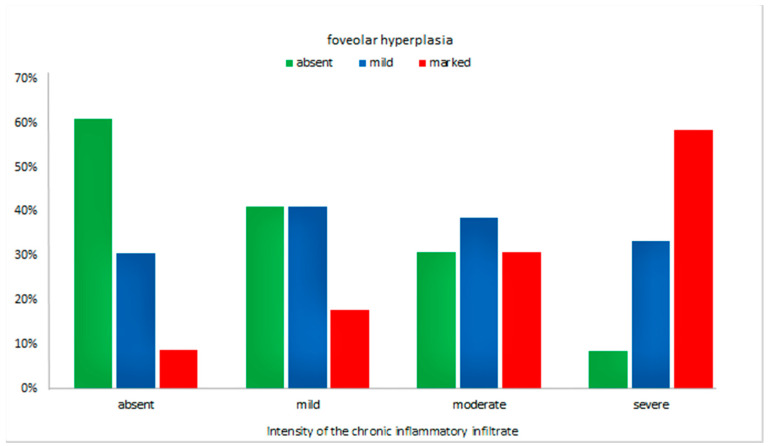
The degree of foveolar hyperplasia, depending on the intensity of the chronic inflammatory infiltrate, irrespective of the *H. pylori* status.

**Figure 3 jcm-09-03639-f003:**
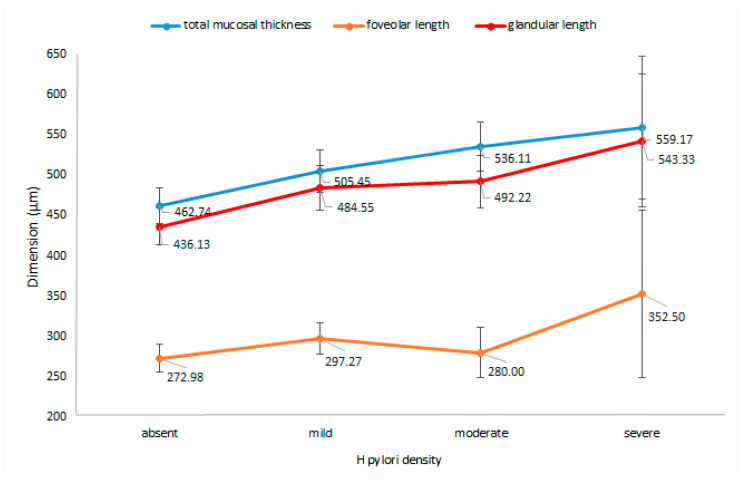
Antral mucosal measurements (total mucosal thickness, foveolar length, glandular length) according to different *H. pylori* colonization grades. Note: Data are expressed as means ± 95% CIs.

**Figure 4 jcm-09-03639-f004:**
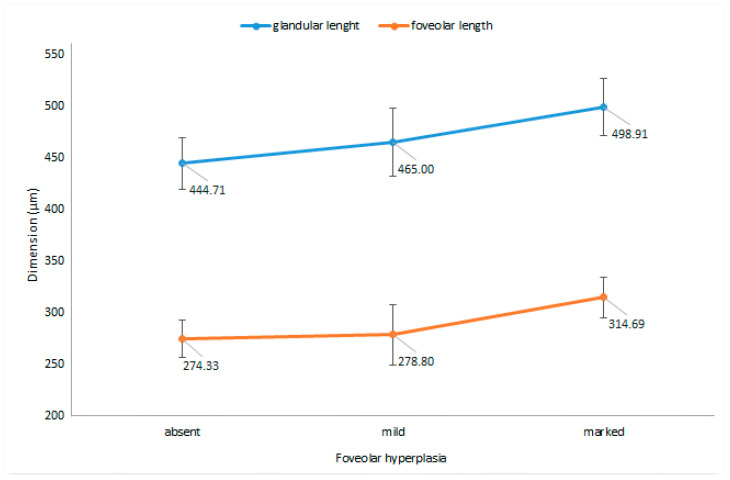
Antral foveolar and glandular length (means ± 95% CIs), depending on the degree of foveolar hyperplasia.

**Figure 5 jcm-09-03639-f005:**
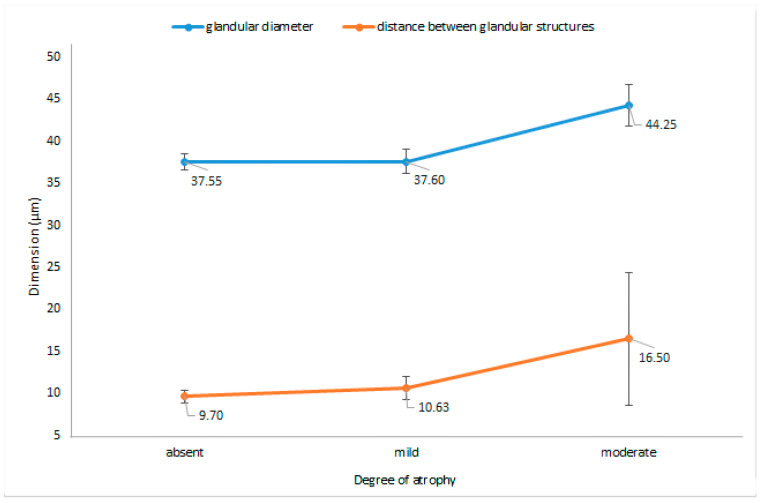
Glandular diameter and distance between the glandular structures in the antral mucosa (means ± 95% CIs) in relation to different degrees of atrophy.

**Figure 6 jcm-09-03639-f006:**
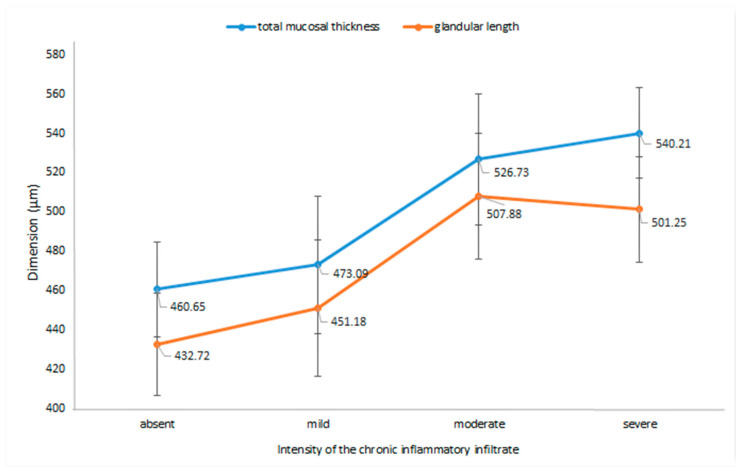
Total mucosal thickness and glandular length of the antral mucosa in relation to the intensity of the chronic inflammatory infiltrate. Note: Means ± 95% CIs are shown for different grades of the inflammatory infiltrate.

**Table 1 jcm-09-03639-t001:** Demographic characteristics of the study group. IQR, interquartile range.

Sex	Number of Patients (%)	Median Age [IQR]	*H. pylori* Positivity (%)
Male	21(32.3%)	12 [9.75–16.25]	10 (29.41%)
Female	44(67.69%)	14.5 [12.5–17]	24 (70.59%)

**Table 2 jcm-09-03639-t002:** Morphologic measurements performed on antral biopsies.

Parameter	Mean/Median (µm)	SD/IQR	Range (µm)
Total mucosal thickness	491.80	±68.37	350–650
Foveolar length	285.84	±52.52	195–455
Glandular length	460	±69.14	265–627.5
Foveolar/glandular length	0.61	±0.08	0.42–0.78
Foveolar length/total mucosal thickness	0.58	±0.07	0.40–0.74
Glandular length/total mucosal thickness	0.95	0.93–0.96	0.68–0.99
Glandular diameter	37.76	±3.57	29–47
Distance between glands	9.5	8–11.5	4–20.5
Glandular cross-sections/40X field	17	±3.45	9.5–26

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
