# Peer review of "Particular Morphological Features in the Diagnosis of Pediatric *Helicobacter pylori* Gastritis: A Morphometry-Based Study"

_jcm, 2020, doi:10.3390/jcm9113639_

Round 1

Reviewer 1 Report

In this study, the authors aimed to investigate the morphometric characteristics of the gastric antral mucosa concerning pediatric H pylori infection. The manuscript is well written and original.

I have some concerns:

  1. Can the authors insert a table in which they describe the morphometric characteristics of children and adults?
  2. In table 1: can we add the presence or the absence of H pylori infection?
  3. In figure 2: the data are related to the H pylori infection?

Reviewer 2 Report

The study has several flaws.

First of all the design is not clear, as well as the objectives. Reported data misses unit of measures. The statistical analysis is not clear, and authors use wrong terms such as "relationships", when they are simply reporting bar graphs. Also, The study population is low. Also, the study misses a control group which is fundamental.

Reviewer 3 Report

In the manuscript titled, “Particular morphological features in the diagnosis of pediatric Helicobacter pylori gastritis- a morphometry-based study”, Ana-Maria Teodora Domșa and co-authors aimed to investigate the morphometric alterations of the antral mucosa in relation to pediatric H pylori infection.

Please refer to comments I have given below.

  1. Some studies support the use of immunohistochemistry, since haematoxylin and eosin staining has been shown to be 42–99% sensitive and 100% specific when compared to immunohistochemistry. Moreover, staining for pylori has a lower inter-observer variation when compared to histochemical stains. Although using immunohistochemistry staining is not routinely indicated nowdays, it seems very relevant for the presented study.
  2. According to the Maastricht V/Florence Consensus Report, for assessment of  pylorigastritis, a minimum standard biopsy setting is two biopsies from the antrum (greater and lesser curvature 3 cm proximal to the pyloric region) and two biopsies from the middle of the body. Therefore, specific details regarding the number of antral as well as corpus biopsies, and the comparison of morphologic findings in different gastric locations should be included in the method and results sections, respectively.
  3. It is well recognized that long-term treatment with PPIs alters the topography of  pylorigastritis. Clinical data regarding PPI use and foveolar hyperplasia assessement in correlation with PPI use is indicated.

Round 2

Reviewer 2 Report

The paper can be accepted in its current form. The authors have edited the previous version according to the reviewers comments.